# iPS cell generation-associated point mutations include many C > T substitutions via different cytosine modification mechanisms

Ryoko Araki [1,2] ✉, Tomo Suga[1,2], Yuko Hoki[1,2], Kaori Imadome[1,2], Misato Sunayama[1,2], Satoshi Kamimura [1,2], Mayumi Fujita [1,2] & Masumi Abe [3] ✉

Genomic aberrations are a critical impediment for the safe medical use of iPSCs and their origin and developmental mechanisms remain unknown. Here we find through WGS analysis of human and mouse iPSC lines that genomic mutations are de novo events and that, in addition to unmodified cytosine base prone to deamination, the DNA methylation sequence CpG represents a significant mutation-prone site. CGI and TSS regions show increased mutations in iPSCs and elevated mutations are observed in retrotransposons, especially in the AluY subfamily. Furthermore, increased cytosine to thymine mutations are observed in differentially methylated regions. These results indicate that in addition to deamination of cytosine, demethylation of methylated cytosine, which plays a central role in genome reprogramming, may act mutagenically during iPSC generation.

iPS cells (iPSCs) hold considerable promise in regenerative medicine and several clinical trials are currently underway[1,2]. For the purposes of transplantation and drug discovery, differentiation systems using iPSCs to generate various cell types have been developed and the use and advancement of mass culture technology and organoids progressed at a dramatic pace. However, researchers and clinicians working in this field remain in the difficult situation where the clinical safety of iPSCs cannot yet be firmly determined with the question of whether genome reprogramming process is accompanied by genomic instability unanswered. The detection of hundreds of mutations per iPSC genome has been reported from multiple groups in both mouse and human lines, as well as in mouse nuclear transfer ES (ntES) cells[3–12]. This phenomenon is of serious concern in terms of possible immunogenicity and, above all, tumorigenicity, and there is an ongoing debate about the reality of this phenomenon and the mechanisms by which these mutations occur[13–15]. One of the most important issues is whether the mutations detected in iPSC genomes are already present in the parental somatic cells (i.e., pre-existing SNVs) or whether they are caused by the iPSC generation processes. Here we find that most detectable mutations in iPSC genomes are de novo and that these genomes contain far more C>T mutations than the germline. In addition, iPSC genomes harbor mutations in regions that are typically not susceptible to these variations, and in specific retrotransposons, namely AluY.

## Results

### Origins of the mutations identified in iPSC genomes

Whether or not genomic mutations occur during the cell lineage conversion from somatic cells to iPSCs is a critically important question to resolve from both a medical and a scientific point of view. There

[1]Stem Cell Biology Team, Institute for Quantum Life Science, National Institutes for Quantum Science and Technology, Chiba, Japan. [2]Department of Radiation Regulatory Science Research, Institute for Radiological Science, National Institutes for Quantum Science and Technology, Chiba, Japan. [3]Institute for Quantum Medical Science, National Institutes for Quantum Science and Technology, Chiba, Japan. ✉e-mail: araki.ryoko@qst.go.jp; abe.masumi@qst.go.jp

are reports that mutations detected in iPSCs are pre-existing single nucleotide variants (SNVs) and were therefore already present in the parent somatic cells, and this debate continues[15]. In this present study, we aimed to obtain a more definitive answer on this issue using three approaches.

We first established iPSCs from the same somatic cells using different methods and the mutations detected in each cell line were then compared. Human dermal fibroblasts derived from a single individual (18year-old female, FC-0024, Lot No. 00967, Lifeline Cell Technology, Frederick, MD, USA) were used as the parent cells and iPSCs were generated using retroviral vectors or integration-free plasmid vectors to introduce reprogramming factors[10,11]. Mutations were identified by WGS analysis of three and five iPSC lines, respectively. The results showed a statistically significant difference in the number of mutations between the methods, i.e. $461.3 \pm 23.1$ for retro virus-mediated vs $710.0 \pm 135.6$ for episomal vector-mediated (Fig. 1a, b and Supplementary Data 1). This result, that the number of mutations is dependent on the gene delivery method, indicates many mutations in iPSCs cannot be explained by pre-existing SNVs.

Normally, WGS results for the parental somatic cell populations used for iPSC generation are employed as a control, or reference sequence (ref seq), for iPSC mutation analysis (Supplementary Fig. 1). However, this ref seq may not include mutations that appeared during the culture of the parental cell population, i.e., mutations that exist in only a few cells in the population, due to the detection limit problem. On the other hand, since iPSCs are derived from a single cell, it has been pointed out that mutations that are missed in the analysis of such parent somatic cell populations can be detected and recognized as mutations unique to iPSC lines[15]. To address this issue herein, we focused on sister iPSC lines established from the same parental somatic cells. Since mutation identification for each sister line was performed independently using the parental somatic ref seq as usual, pre-existing SNVs that were missed may be detected in sister iPSC lines and, depending on the developmental relationship between them, may be detected as common mutations between sister cell lines (i.e., shared SNVs), and this possibility increases as the number of sister lines increases.

We therefore examined the 35 sister lines we had previously established using the same parental cell population for the presence of shared SNVs. The iPS cell lines were: 33 iPS cell lines and 2 ntESC lines independently established from one type of MEF (MEF5 prepared from a single embryo) (Fig. 1c, d). To our knowledge, no comparative study of as many as 35 lines of iPS cells established from the same parental somatic cells has been reported, providing crucial information in the discussion of pre-existing SNVs.

The number of shared SNVs detected among these sister strains is highlighted in Fig. 1c in blue (e.g. there was one shared SNV between R3F-60 and NAC1-12). WGS analysis revealed a total of 25,966 SNVs (mean, $741.9 \pm 320.0$ per cell line) identified across 35 lines, of which only 24 (0.28%) were shared. There were six cell lines (539, 19004, R3F-70, 1170, NAC1-7, NAC1-11) in which no shared SNVs were found, several lines in which only one was detected, one pair in which four were detected (594 vs. 866), and two pairs in which five were detected (Thy-48 vs. 420 or 427). Even with the extreme cases, there were only one pair in which 11 shared SNVs was found (420 vs. 427; Fig. 1c).

Among the mutations that arise in somatic cell populations during culture, those that occur at an earlier stage should be excluded by using the ref seq data of the parental somatic cell population, since they are expected to form larger clusters within the population. Notably in this regard, even the PCR-based ultradeep sequencing test of parental cells, which can detect mutations in 1/2500–1/5000 cells within a parental somatic cell population[10,16], has not detected the mutations detected in each iPSC by our WGS mutation analysis using the parental cell ref seq data. Hence, the lack of shared SNVs even in the 35 sister clones we analyzed by WGS indicates that most of the

mutations detected in these 35 iPSC clones, averaging $741.9 \pm 320.0$ cells/line, were not pre-existing but arose intensively over a short period of time in the final phase of the culture period of parent cell population, including cell lineage conversion step into iPSCs.

As mentioned above, standard mutation analysis of iPSCs uses the WGS data of the parent somatic cell genome as the reference sequences. However, this may cause a sensitivity problem. Hence, we further performed mutation analysis in our present study using sister iPSC lines as well as parental somatic cell for ref seq data and compared the mutations between the two. If there were many different SNVs in each cell of the parent somatic cell population, and if these variations were detected as iPS clone-specific mutations, then the ref seq information of one of the sister lines instead of the parent somatic cell should miss significantly more mutations than the ref seq data of the parental cells. This is because the sister line is of single cell origin and is therefore only a subset of the parental somatic cell population. On the other hand, the sister line WGS data should detect SNVs at very low frequencies that are not detected in the parental cell ref seq due to sensitivity issues. The number and content of mutations detected will be expected therefore to differ between the parental cell ref seq and the sister line WGS data. Furthermore, if the data from the different sister cell lines are used as the ref seq, the results should also differ.

In our specific analyses, five independent sister clones established using somatic cells from the same single individual were used[11]. The mutations identified in each sister clone were compared with those in the parent somatic cells and with each of the other four different sister clones as references (Fig. 1e). This analysis resulted in a total of 25 (5 clones × 5 different ref seq = 25) sets of mutation analyses (Fig. 1e, upper right). The results indicated almost no differences in the outcomes of using the parent somatic cell and any of the sister clones as the reference sequences (99.0–100% concordance rate of SNVs detected; Fig. 1e). This finding is very surprising in two ways. First, the sister iPSC ref seq data, all of which are derived from a single cell, contained all the information contained in the ref seq derived from the parent somatic cell, which is a population. Second, the ref seq data for the parental cell population, in which each cell constituting the population should have many different mutations, contained the same information as the single-cell-derived iPSC-derived ref seq information, i.e., no sensitivity limitation issues were observed. These results thus indicated that the parent somatic cell population and its five independently established sister strains harbored nearly identical information as a ref seq, such that there was no heterogeneity in the parent somatic cell population genome that could explain the hundreds of variations found in iPSCs[5].

The results of these present analyses indicate that pre-existing SNVs are present in some cells in the iPSC parent somatic cell population. Even with this apparent heterogeneity in the parental cell population in terms of mutations, its effect on correct mutation identification in iPSCs will be very low as long as bioinformatics processing using ref seq information of the parental cell population is rigorously performed.

## Variations of up to 100-fold exist in the SNV numbers among human iPS cell lines

Since we confirmed that the mutations observed in iPSCs are de novo and that sister lines can also be used to generate ref seq information, we next employed our analysis system to conduct mutation analysis of 75 human iPSC lines, 53 for which WGS information on sister clones is also available in public databases, along with the WGS results of their parent somatic cells (https://www.hipsci.org/)[17], and 22 of which were generated in our laboratory (Fig. 2a, Supplementary Table 1 and Supplementary Data 1).

We found that 38 (50.7%) of the 75 iPSC lines analyzed had more than 1000 mutations, including 5 lines with more than 5000 mutations and 2 lines with more than 10,000 mutations, the greatest amount

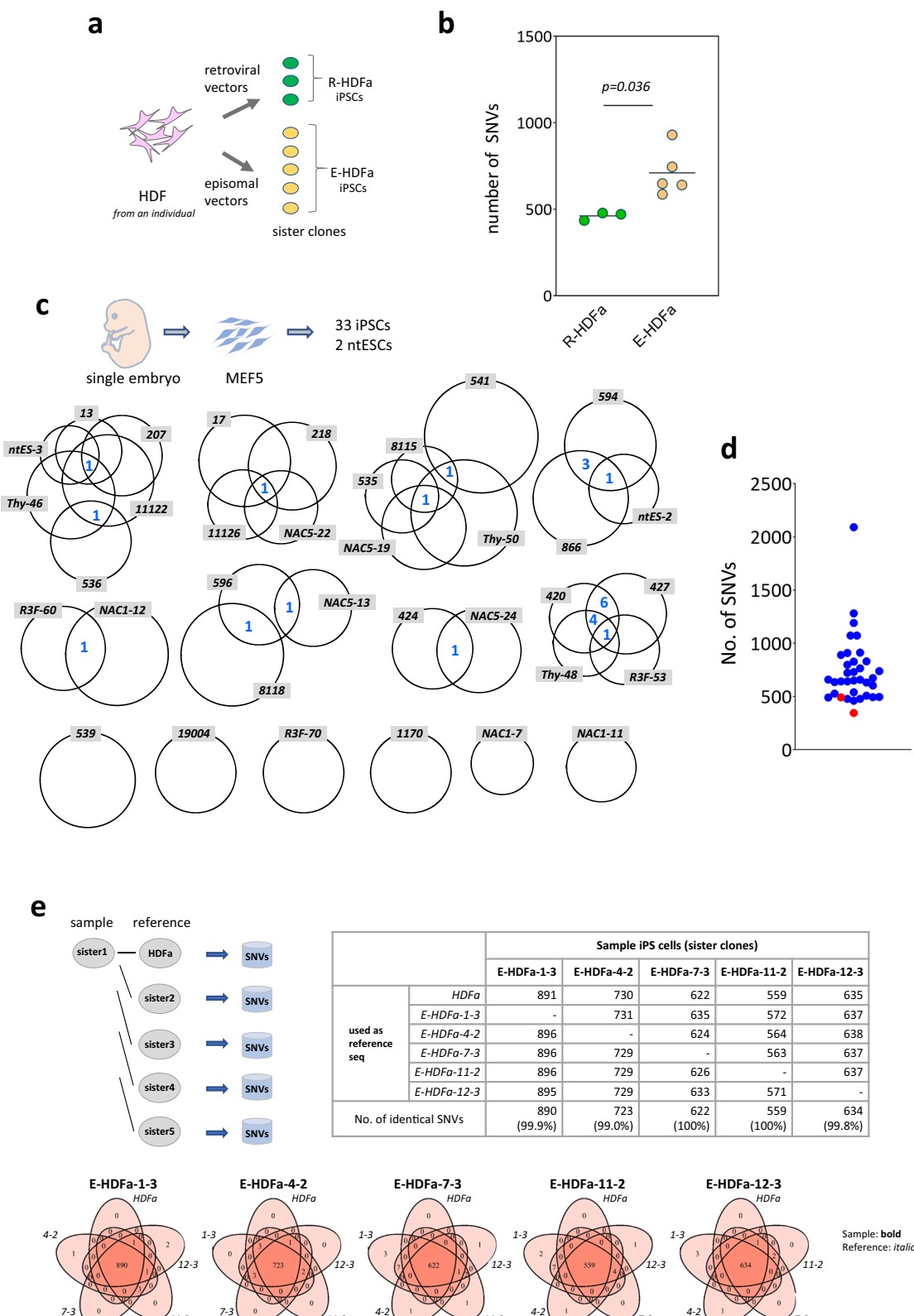

being 12,857 (Fig. 2a). The cell lines with more than 10,000 SNVs would be expected to have more than 100 coding sequence (CDS) mutations and had 111 and 106 such mutations indeed (Supplementary Data 1). Notably also, the 53 clones established outside of our laboratory using the Sendai virus vector were found to have particularly large numbers of mutations, while there were also large differences in the number of

mutations between lines (Fig. 2a). Not only did the frequency of mutations vary by generation procedure ($102.1 \pm 21.2$, $710 \pm 135.6$, $461.3 \pm 23.1$, and $2382.8 \pm 2432.2$ mutations for CB-epi, E-HDFa, R-HDFa, and HipSci, respectively; Supplementary Table 1)[10], there were also large differences in the number of mutations among the sister cell lines (Supplementary Fig. 2a). On the other hand, there was no positive

**Fig. 1 | De novo mutations. a** Experimental design Three iPSC lines were established by retroviral vectors, and five iPSC lines were established by episomal vectors from the same human somatic cells. **b** Point mutation frequencies identified in iPSCs established from the same HDFs. Total numbers of SNVs in each sister clone are shown. R-HDFa, iPSC lines established by retroviral vectors (*n* = 3 biological replicates); E-HDFa, iPSC lines established by episomal vectors (*n* = 5 biological replicates). For statistical analysis, the two-tailed *t*-test was employed. SNVs and positions of each clone are shown in Supplementary Data 1. **c** Analysis of 33 iPSCs and 2 ntESCs established from the same somatic cells. The size of the circles indicates the number of SNVs and the numbers of mutations common among clones

(shared SNVs) are denoted in blue in the Venn diagram. Thirty-three iPS cell lines established using retroviral method and 2 ntES cell lines via a nuclear transfer method from the single embryo-derived fibroblasts (MEF5) were investigated. **d** Point mutation frequencies detected in each iPS and ntES cell clone. Total numbers of SNVs in each clone are shown. Blue, iPSCs; red, ntESCs. **e** Detection of SNVs when a sister iPS clone is used as the reference sequence. Upper left: Schema for the analysis. Upper right: Number of SNVs. SNVs identified when parent somatic HDFa or each of the 4 sister clones was used as reference sequence. Lower: Relationship between SNVs identified using each of the five different reference sequences. Source data are provided as a Source Data file.

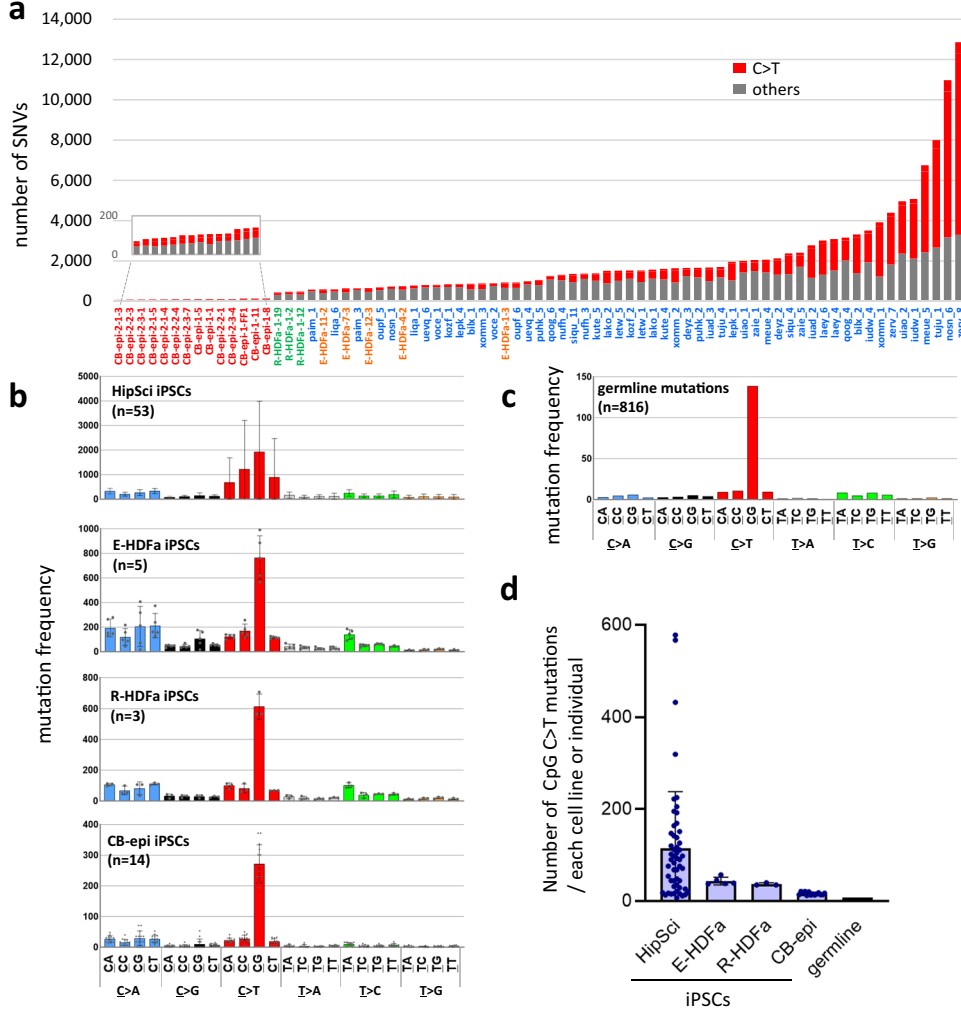

**Fig. 2 | C to T mutations in iPS cells. a** Variation in the number of point mutations in human iPS cell lines. The X-axis shows the names of the iPSC lines. Red: 14 iPSCs established from cord blood using all-in one episomal vectors (CB-epi iPSCs); green: 3 iPSCs established from HDF using retrovirus vectors (R-HDFa-iPSCs); orange: 5 iPSCs established from HDF with episomal vectors (E-HDFa-iPSCs); and blue: 53 iPSCs established from HDF with Sendai virus vectors (HipSci iPSCs). The total numbers of SNVs are shown on the Y-axis. C > T transitions are denoted in red. **b** Mutational profiles of human iPSCs. Mutant bases were classified in 24 different ways in accordance with their 3′-neighbor bases. The Y-axis indicates the numbers

of mutations normalized by the genomic abundance of their two contiguous bases, shown as number per $10^9$ bases. Error bars show the SD of the mean. **c** Profile of germline mutations. **d** Number of C > T mutations on CpG sequences. Numbers of C > T mutations at CpG sequences are shown as the number of mutations per cell line or per individual. HipSci iPSC (*n* = 53 biological replicates); E-HDFa-iPSC (*n* = 5 biological replicates); R-HDFa-iPSCs (*n* = 3 biological replicates); CB-epi iPSCs (*n* = 14 biological replicates). Error bars show the SD of the mean. Source data are provided as a Source Data file.

correlation between mutation frequency and the age of the individual from whom the parent somatic cells were derived (Supplementary Fig. 2b). Hence, our mutations results for the 75 lines tested in this analysis also supported that iPSC mutations are not pre-existing SNVs. It should be noted that the results of this analysis do not necessarily indicate that the use of Sendai virus vectors for iPSC generation causes

more mutations. It is necessary to consider other factors in these lines since large differences were observed between clones established by this method.

To further explore the characteristics of mutations detected in iPSC genomes, we conducted mutation signature analysis and normalized the results by the number of each sequence present in the

genome. The results revealed that CpG is a prominent mutation-prone site, with a resulting C to T transition efficiency that is more than 10-fold higher than other low-frequency sites, with the same trend observed in both human and mouse iPSC lines (Fig. 2b and Supplementary Fig. 3). Although mutations in CpG sequences are well known polymorphisms among individuals (Fig. 2c), the frequency in iPSCs was found to be higher than that among individuals (approximately 2.0- to 14.0-fold; Fig. 2d). Furthermore, in each iPSC line, C to T transitions at a CpG site (hereafter denoted as CpG C > T) correlated with the total number of mutations detected (Supplementary Fig. 4). The intragenomic distribution of mutations also presented a unique profile in iPSCs. CpG C > T mutations in the germline neatly reflects the distribution of CpG sequences within the genome, whereas those in iPSCs showed a bias toward intergenic regions (Fig. 3a). More interestingly, the amount of CpG C > T mutations, which was increased in iPSCs, showed elevated numbers arose in retrotransposons, especially in short interspersed nuclear elements (SINEs) (Fig. 3b and Supplementary Table 2). On the other hand, no CpG specific elevation of C > T mutations was observed in long interspersed nuclear elements (LINEs) and long terminal repeat (LTR)-type retrotransposons. Most interestingly, among the SINE subfamilies, one type of subfamily, AluY that is the most active[18], showed a large increase of CpG C > T mutations in iPS cells (Fig. 3c, d). Considering that the C to T mutations in non-CpG sites do not show an increased mutation frequency in SINEs in iPS cells, our results seem to suggest that CpG-specific mutations in SINEs in iPS cells are caused by a different molecular mechanism than the previously known deamination of cytosine (Fig. 3b).

### The C to T transition mutation profiles of iPS cell genomes differ from those accumulated in germlines as inter-individual differences: relationship between DNA demethylation and mutation

The existence of iPSCs with more than 10,000 de novo SNVs was demonstrated in this present study, and a large number of C > T transitions were revealed among them (Fig. 2a). The number of C > T mutations showed a positive correlation with the total number of mutations. However, when the cytosine-containing sequences in which mutations occurred were divided into CpG and non-CpG, a contrasting correlation emerged i.e., the percentage of C > T transitions at non-CpG sites correlated positively with the total number of SNVs, whereas the percentage of mutations occurring at CpG sites was found to be inversely correlated (Fig. 4a, b). Given that the C > T transition at non-CpG sites is thought to be due to deamination, this finding also indicated that a mechanism other than this was involved in generating the C > T transition at CpG sites in the iPSCs with low total mutation numbers and with a small percentage of these transitions at non-CpG sites. On the other hand, the significant increase of C > T mutations at non-CpG sites in iPSCs was also particularly noteworthy. The ratio of C > T changes at non-CpG to CpG sites appeared to be different in iPSCs from that in the germline (Supplementary Fig. 5). Although the mutation profile in the germline was consistent with that in previous reports, i.e. the rate of 5mC to T in the germline was reported previously to be about four times higher than the rate of C to U deamination[19–21], non-CpG C > T mutations were significantly increased in our iPSCs, depending on the derivation method. The aforementioned ratio was thereby completely reversed in HipSci iPSCs compared to the germline. This observation indicates a C > T mutagenic system that is unique to iPSCs and radically different from the germline.

A possible CpG-specific reaction other than demamination is the methylation of DNA. Gene expression analysis during the cell lineage conversion from somatic cell to iPSC has confirmed the induction of *Tet1* and *Tet2*, enzymes thought to be responsible for DNA demethylation, during iPSC formation[22–25]. We then conducted forced expression experiments with *Tet1* in iPSC generation (OSKC + Tet1) (Fig. 5a–c). As in previous studies, an increase in iPSC generation

frequency was observed upon introduction of *Tet1*[25](Fig. 5d). Subsequent WGS analysis of six Tet1-iPSC lines and three control lines revealed an increase in CpG sequence-specific C to T transition frequency upon *Tet1* transducion (Fig. 5e, f). In addition, we also observed an increase in mutations in SINEs (Fig. 5g). This observation seemed to reproduce the phenomenon observed in the mutation analysis of human iPSCs genome, i.e. increased mutations in AluY retrotransposons (Fig. 3c).

Hence, although *Tet1* gene transduction experiments have suggested that DNA demethylation during iPSC generation has mutagenic potential, further analysis is needed to confirm that the C > T mutations detected in iPSC genomes are indeed due to DNA demethylation. Since DNA demethylation mediated by the Tet products is a key reaction in genome reprogramming and occurs on a large scale, it is crucial to determine whether this process is mutagenic or not. One direct way to confirm this idea would be through Tet suppression. However, it has been reported that suppression of the Tet family inhibits genome reprogramming itself[24,25]. Here, we also performed suppression experiments using shTet1 and shTet2, but iPSC generation itself was inhibited (Supplementary Fig. 6a). Moreover, the iPSCs that were generated under these conditions showed an incomplete morphology, especially at the beginning of their generation and SNV analysis of these imperfect iPSCs showed no difference from controls (Supplementary Fig. 6b).

This situation required an approach other than reverse genetics. We therefore investigated the findings that have strongly suggested a close relationship between DNA demethylation and CpG C > T mutations. Further analysis of the results shown in Fig. 4 confirmed the contrasting correlation i.e., the percentage of C to T transitions at non-CpG sites showed a strong positive correlation (r = 0.82) with the total number of SNVs, whereas the percentage of mutations occurring at CpG sites showed a weak negative correlation (r = −0.21) (Fig. 6a, b). Interestingly, it was further noted that in iPS lines with significantly lower total SNV counts, the ratio of CpG C > T to all mutations was significantly higher (Fig. 6a).

Given that non-CpG C > T transitions are thought to be due to deamination, we considered that a mechanism other than deamination might be operating in iPSCs with such a low total number of mutations and in which C > T transitions at CpG sites are common, but at non-CpG sites are rare. We focused our analysis therefore on iPSCs with few mutations, especially those derived from erythroblasts and HDFs. We compared the mutation rates at DMR (differentially methylated region) sites[26] and other genomic regions and found that CpG C > T mutation at DMRs occurred more frequently in erythroblast-derived iPSCs (Fig. 6c). A similar trend was clearly observed in HDF-derived iPSCs, which are the second least mutated iPSCs after erythroblast-derived iPSCs, although it was weaker than in the erythroblast-derived iPSCs. In addition, the expression of the *TET1* gene in their parental cells showed a similar trend, i.e., the higher its expression level, the more likely the cells were to have CpG C > T mutations in the DMR (Fig. 6d). By contrast, no similar trend was observed in mature iPSCs i.e., between erythroblast-derived and HDF-derived iPSCs, or between *TET1* gene expression and the number of CpG C > Ts in HipSci iPSCs (Supplementary Fig. 7a, b). To further assess this relationship, analyses of the very beginning of the cell lineage conversion process (day 2-3), when a large-scale reorganization of genome methylation pattern is expected to take place, would also be necessary.

Further to this, the marked elevation of the C > T mutation frequency at CpG sites in SINEs (most notably in evolutionarily young families i.e. AluY families) in iPSCs, which is not seen at non-CpG sites, also appears to suggest a close relationship between DNA demethylation and this mutation (Fig. 3b–d). Indeed, upregulation of the expression of AluY families in iPSCs compared to their parental cells was also clearly demonstrated (Supplementary Fig. 8). Furthermore,

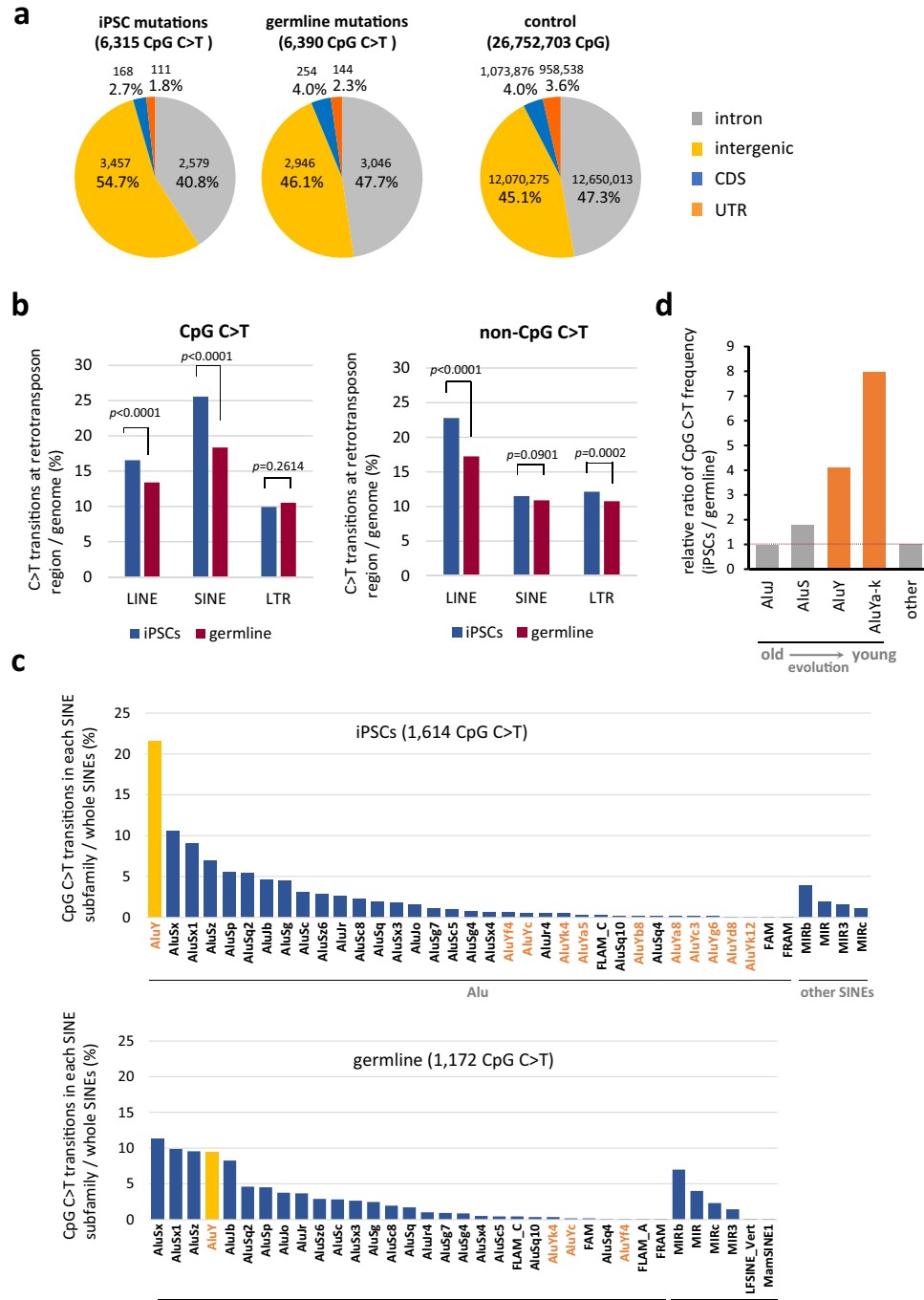

**Fig. 3 | Frequent CpG C > T transitions in retrotransposons in iPSCs.**
**a** Distribution of CpG C > T mutations in the genome. Distribution of CpG sequence in the genome is shown in 'control'[55,56]. **b** Mutations at CpG or at non-CpG on retrotransposon sequences. Percentages of mutation positions on LINE, SINE and LTR-retrotransposons are shown. For statistical analysis, the two-tailed Z-test was employed. See Supplementary Table 2 also. **c** CpG C > T mutations in each SINE subfamily. The SINE subfamily names are shown on the x-axis. AluY and other AluY subfamilies, Ya through Yk, are denoted in orange. The y-axis indicates the percentage of CpG C > T mutations detected in each SINE subfamily of the total

number of CpG C > Ts detected within whole SINEs. One SINE subfamily, AluY, which showed a marked increase in the number of mutations in iPSCs, is shown in yellow. **d** Ratio of the CpG C > T frequency of each SINE subfamily between iPSCs and germline. The iPSC to germline CpG C > T ratio in each subfamily was calculated using the % data shown in Fig. 3c, and the values are shown relative to AluJ as 1. AluJ and AluS are the sum of the values of the subfamilies belonging to the AluJ and AluS families, respectively, and AluYa-k is the sum of the values of the subfamilies, AluYa to AluYk. Shown in orange are the evolutionarily young subfamilies AluY and AluYa-k. Source data are provided as a Source Data file.

the degree of upregulation differed between iPSC clones for almost exclusively AluY families (shown in orange), but not for the non-AluY retrotransposons (gray). This observation that the degree of expression induction varies with each iPSC generation event suggested a profound relationship between iPSC generation and transcriptional

induction of AluY (Supplementary Fig. 8). Increased retrotransposition activity in iPSCs has recently been reported[27], and DNA demethylation, followed by transcription, is the first step in this reaction.

These observations suggest that during genome reprogramming, DNA demethylation reactions are closely related to CpG C > T

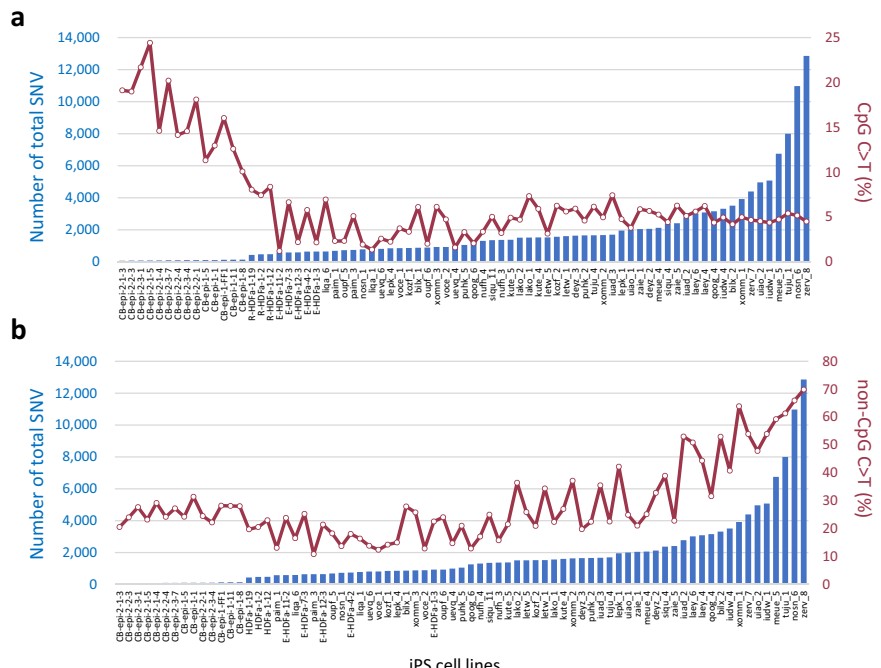

**Fig. 4 | Relationship between total number of mutations and C to T transition (%) (at CpG vs. non-CpG sites).** The X-axis shows the names of the iPSC lines. The total number of SNVs and the percentage of CpG C > T mutations (**a**) or non-CpG C > T mutations (**b**) are shown on the Y-axis. Source data are provided as a Source Data file.

mutations. Because of the biological significance of this, further analysis and more direct evidence is needed in the future.

Furthermore, elucidating the relationship between DNA demethylation and CpG C > T mutations in biological phenomena other than genome reprogramming will be an important issue that may provide insights into the mechanisms of carcinogenesis and other diseases. Therefore, we also analyzed the effect of forced expression of *Tet1* on mutagenesis in somatic cells that are not undergoing genome reprogramming. However, WGS analysis of a *Tet1*-transduced somatic cell population cannot provide valid data because this cell population is very heterogeneous in terms of mutations. To accurately study the effect of the *Tet1* gene on mutagenesis therefore, it will be necessary to isolate a single transgenic cell from a post-transfection cell population, and this requires a system that can culture a single cell to a sufficient population size.

To date single cell expansion from primary cell populations remain not possible in mice and very few human cell types are capable of expansion from single cells to WGS-enabling levels. We therefore used p53KO-MEFs (5 and 16) to compare *Tet1*-transduced and untransduced (control) cells (Supplementary Fig. 9a, b). Briefly, the cells were infected with a retroviral vector containing *Tet1* gene, and single cell isolation was performed three days later. The cells were then allowed to grow for one month (and some were set aside for WGS). After this, single cell isolation was performed again, and the number of cells was allowed to grow to 500–700 cell populations and subjected to WGS. In this system therefore, *Tet1* was expressed intracellularly for one month to observe its effect. As a result, a trend toward increased CpG C > T mutations was observed in *Tet1*-transduced cells, although this was not statistically significant (Supplementary Fig. 9c). These observations suggested that mutations may also occur during biological events other than genome reprogramming, such as carcinogenesis following deletion of the p53 gene, and regeneration events in amphibians, where a transient decrease in p53 activity occurs[28]. This sheds new light on the molecular mechanisms underlying these important biological phenomena. In our present experiments, we could only use p53−/− cells due to technical limitations, but addressing the question of whether DNA demethylation produces mutations even

under normal p53 conditions is definitely critical, as it may be strongly associated with development and differentiation.

## Biological relevance of the C to T transition at CpG sites

Although we observed dramatically less mutated iPSCs than reported previously, i.e., erythroblast-derived iPSCs (about 1/5 of conventional iPSCs), even these cell lines contained several-fold more mutations than ES cells[7,10]. Detailed analysis revealed that the C to T transition occurred at a significantly higher frequency in the iPSCs, especially at CpG sites. The frequency of this mutation was about 10-fold higher than similar mutations observed in germlines (Fig. 2d). We therefore discussed the biological effects of a C > T single nucleotide substitution occurring at a CpG site and examined each of the C > T mutations detected in the iPSC genomes in this present study. Four possible mechanisms of biological effects were considered as follows: 1) mutations in the coding sequences (CDS); 2) mutations affecting splicing; 3) mutations in genomic regions other than the CDS, such as intronic and intergenic regions; and 4) mutations in the CpG island (CGI). In the case of 2), no similar instances were found among the mutations detected in the iPSCs, but in the other three types of mutations were confirmed in this study.

With regards to CDS mutations described in 1) above, a total of 168 mutations were detected in the CDS of iPSC genomes in a nonsense: missense: silent ration of 8: 97: 63, of which 58 (34.5%) were substitutions of the basic amino acid arginine (Arg)(Supplementary Fig. 10a–c and Supplementary Data 2). Furthermore, eight nonsense mutations were detected that, without exception, involved an Arg (CGA) to stop codon (TGA) transition (Fig. 7a). Charged amino acid substitutions were also characteristic. The acidic amino acid glutamine (Glu) was substituted with a basic amino acid lysine (Lys) in all 11 locations where it was detected, which can cause dramatic changes in its function, such as complete inactivation. Mutations of proline (Pro) residues, which fix the protein structure, were detected at 14 sites, 11 of which (78.6%) were silent, but 3 of which involved a leucine (Leu) substitution. Pro to Leu mutations of INS gene have been reported to cause diabetes[29]. The genes containing the CpG mutations detected in this study included three that have been previously reported to be

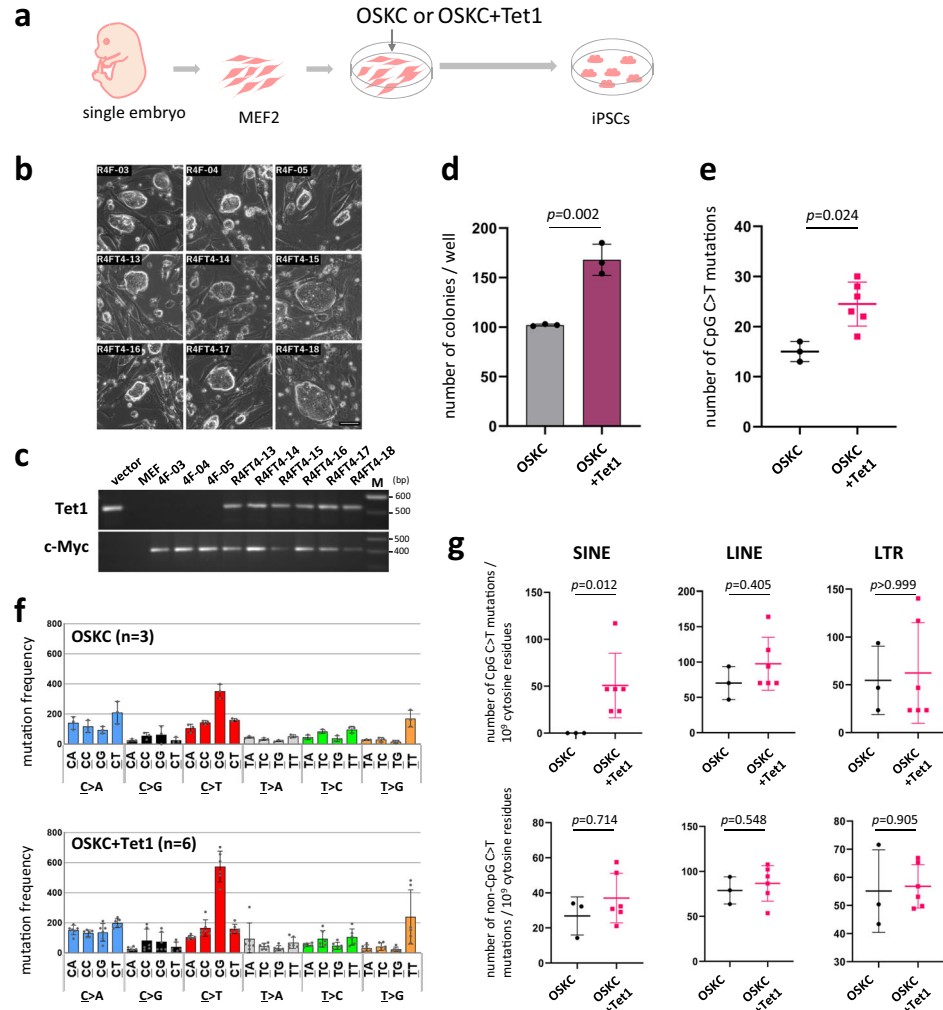

**Fig. 5 | Effects of Tet1 on C to T mutation in iPSCs. a** Schema for iPS generation with Tet1 forced expression (**b**) Tet1-iPSC colonies. Tet1-iPSCs: R4FT4-13, 14, 15, 16, 17 and 18. iPSCs (control): R4F- 03, 04 and 05. Scale bar, 50 μm. **c** Confirmation of retroviral vector integration. Integration of exogenous *Tet1* and *c-Myc* were confirmed by PCR. **d** iPSC generation frequency. The colony formation rate was analyzed using Nanog-GFP MEFs, and GFP positive colonies were counted (*n* = 3 technical replicates). Error bars show the SD of the mean. Statistical analysis was performed using two-tailed t-test for three wells each. **e** Increased CpG C > T mutations frequency due to forced expression of Tet1 during iPSC generation. The

number of C > T mutations at CpG sequences is shown. OSKC iPSC lines (*n* = 3 biological replicates); OSKC+Tet1 iPSC lines (*n* = 6 biological replicates). Error bars show the SD of the mean. The two-tailed Mann−Whitney-U test was used for statistical analysis. **f** Mutation profiles of iPSCs (**g**) CpG C > T and non-CpG C > T mutations in retrotransposon regions. OSKC iPSC lines (*n* = 3 biological replicates); OSKC+Tet1 iPSC lines (*n* = 6 biological replicates). Error bars show the SD of the mean. The two-tailed Mann−Whitney-U test was used for statistical analysis. n.s. not significant Source data are provided as a Source Data file.

associated with cancer: BRIP1, SETD1B, and TERT (Supplementary Fig. 10d) (https://cancer.sanger.ac.uk/census)[30].

Second, the C > T transitions at CpG sites to splicing in the peri-exon region is also known to have many effects, causing dramatic conformational changes in proteins such as deletion and exon skipping[31]. For the situation in 3), the critical question is whether CpG C > T mutations in genomic regions other than coding regions are associated with biological aberrations. A large number of such mutations have been reported as SNPs and are powerful tools for identifying genes or regions highly associated with disease (i.e. disease-associated SNPs). In recent years, attempts have also been made to search for causal SNPs in disease. Alsheikh et al. conducted a comprehensive literature review of non-coding SNPs and reported that 309 causal SNPs have been experimentally validated[32]. These included 52 individual CpG C > T mutations, many of which are in transcriptional regulatory regions and are associated with reduced gene expression. Some have even been shown to have altered transcription factor binding. In addition, CpG C > T mutations have been shown to be risk

factors for type 2 diabetes, Parkinson's disease, ischemic heart disease, etc. (Supplementary Table 3). Indeed, among the mutations detected in iPSCs in this present study, we detected a close relationship between the mutation and methylation profile. We used sister clones to compare cells with similar methylation backgrounds to study the effect of the CpG C > T mutation on methylation. As a result, an interesting pair of sister iPSC clones (nosn_1 and nosn_6) was detected; in this pair, only nosn_6, which has the CpG C > T mutation, showed decreased methylation levels around this mutation. However, this result does not necessarily indicate that this mutation affects methylation. Therefore, we further examined the methylation levels in this region in the remaining 24 sister clone pairs and found that even in a small number of pairs (2/24 pairs), there were large differences in methylation levels between the sisters (all five hypomethylated CpG sites around the mutation detected in nosn_6 had a β value of 0.2 or greater). This result indicates that this region can be methylation variable even without the identified CpG C > T mutation, suggesting that the mutation is a consequence rather than a cause of the hypomethylation. Notably, the

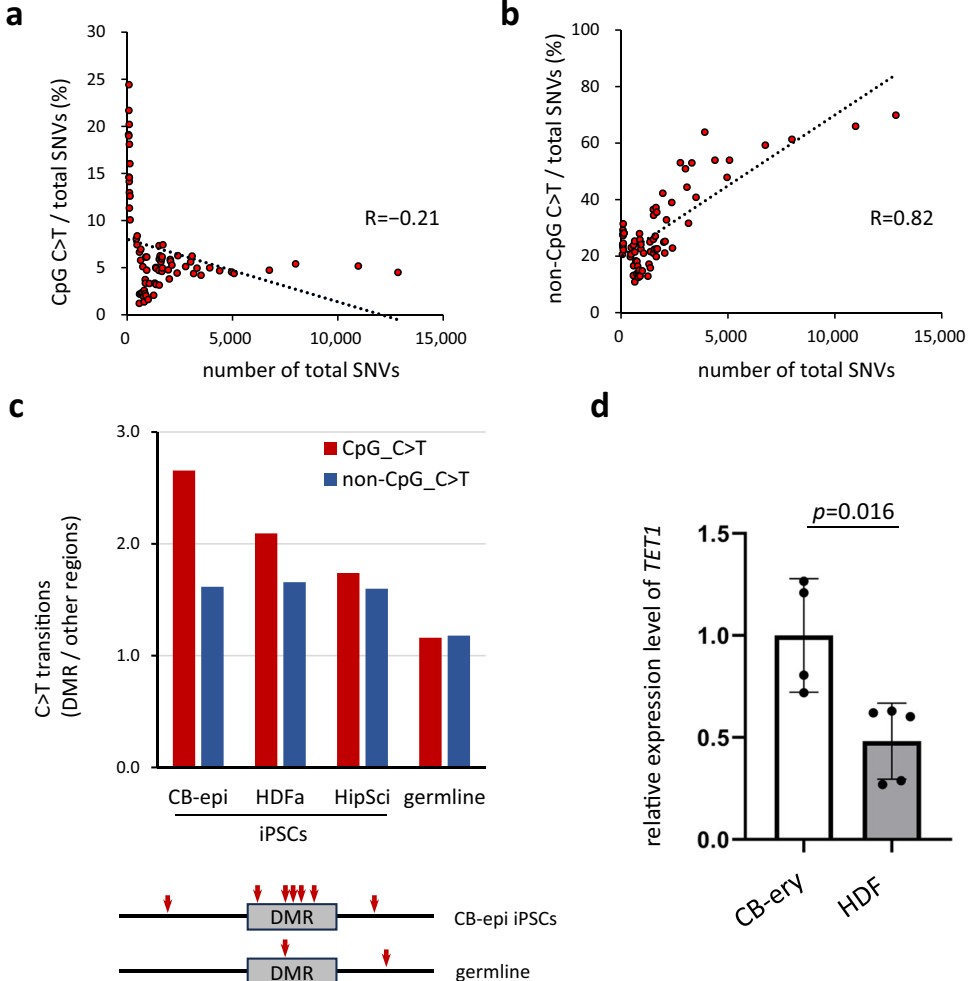

**Fig. 6 | Close relationship between DNA demethylation and CpG C > T in Erythroblasts-derived iPSCs. a, b** Correlation between the percentage of C > T mutations and the total number of SNVs. Human iPSC lines shown in Fig. 2a are analyzed (*n* = 75 biological replicates). The percentages of CpG C > T mutations (**a**) or non-CpG C > T mutations (**b**) are indicated on the Y-axis. Linear approximation was performed and the correlation coefficients are shown. **c** High frequency of CpG C > T transitions in DMRs of CB-epi iPSCs. C to T transitions in DMR in various iPSCs are shown. The Y-axis indicates the ratio of C > T transitions in these DMRs to those in other regions. Red arrows indicate CpG C > T transitions. **d** *TET1* expression level in the parental somatic cells. *TET1* mRNA expression was determined by RT-qPCR. The data were normalized to *GAPDH* expression and the average *TET1* expression in CB-ery was designated as 1. CB-ery, erythroblast-rich fractions expanded from the cord blood (*n* = 4 biological replicates, 4 independent donors); HDF human dermal fibroblasts (*n* = 5 biological replicates, 5 independent donors). Error bars show the SD of the mean. The two-tailed Mann–Whitney-U test was used for statistical analysis. Source data are provided as a Source Data file.

difference in methylation between the sisters is greatest between nosn_1 and nosn_6, so the effect of CpG C > T in such a region prone to methylation fluctuations is a question worth clarifying in the future (Fig. 7b, c). Thus, a C to T transition occurring at a single CpG site, even outside the coding region, has the potential to cause abnormalities. Finally, with regard to 4), many CpG C > T mutations detected in iPSCs were located around a TSS region, where mutations are usually rare. There is also a tendency for mutations to occur less frequently closer to the TSS region, which was not observed in iPSCs (Fig. 7d, e).

## Discussion

It has long been disputed whether the mutations detected in iPSCs are de novo or pre-existing SNVs from the parental somatic cells. The most direct and convincing observation in this regard has always been to perform a WGS observation of the genome of one of the parental somatic cells. However, it is not possible to perform point mutation analysis of a single cell genome with high accuracy due to current technical limitations[33]. Therefore, it became necessary to grow single cells in culture to a population size for which WGS is possible. In this study we introduced a marker gene into a single cell of the parental cell

population and attempted to propagate it among the other cells. However, the introduction of the gene into primary cultured cells was in itself a major stress on the cells, and there was almost no growth after the introduction. In addition to fluorescent marker genes, we also attempted to introduce several others such as antibiotic resistance genes. We utilized the nanopipette method which made it possible to introduce the gene into targeted cells under a microscope and to confirm and follow-up on the introduction of the gene. However, it was confirmed that primary cells injected by this method almost cease proliferating within the cell population (Supplementary Fig. 11). In other words, compared to cell lines, primary cells are much more sensitive to gene transfer and typically undergo growth arrest. Direct analysis of each individual parental somatic cell was therefore not possible, and we thus used the three non-direct approaches reported herein.

Here we identified human iPSC lines harboring more than 10,000-point mutations (Fig. 2a). In these instances of more than 10,000 mutations in iPSCs, there were more than 100 CDS mutations identified (Supplementary Data 1). Furthermore, in iPSCs with such a large number of mutations, C to T mutations in sequences other than CpG

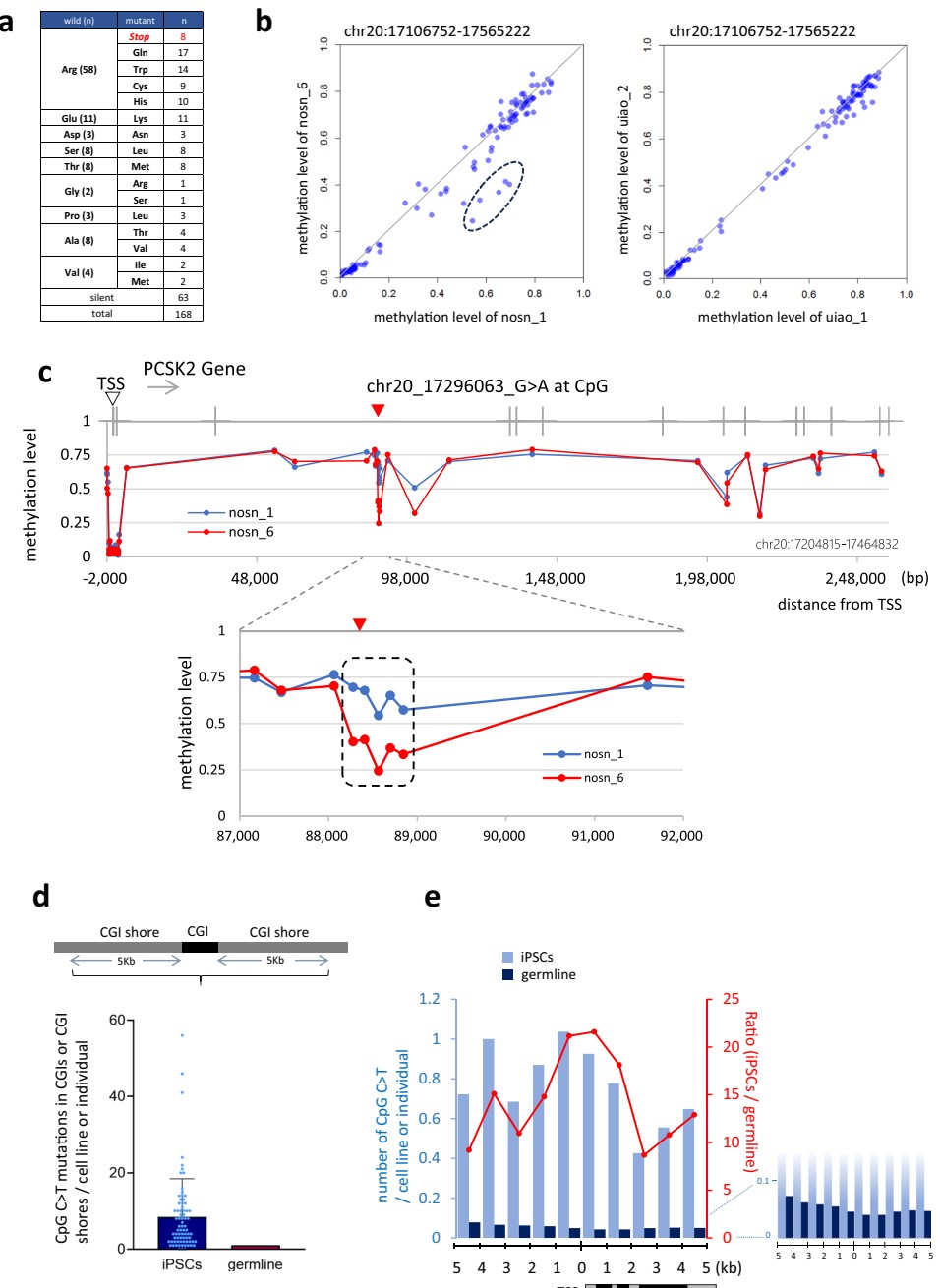

**Fig. 7 | Influence of CpG C>T mutations. a** CpG C > T mutations in CDS. Amino acid substitutions resulting from CpG C > T mutations in iPSCs are shown. **b**, **c** A differential methylation region found adjacent to a CpG C > T transition detected in sister clones analysis. Each dot indicates the methylation value (beta value) measured by Illumina 450 K methylation analysis (HipSci project). **b** The mutation (chr20: 17296063, nosn_6) was detected in a iPSC, clone nosn_6, which was not detected its sister iPSC, clone nosn_1. This C > T site is located in the intronic region of the PCSK2 gene. DNA methylation levels within the ~460 kb genomic region (PCSK2 gene region ±100 kb, chr20:17106752-17565222) between sister clones are plotted and that of another iPS set without mutations in this region (uiao_1 and uiao_2) is also shown as a control. The five hypomethylation sites in nosn_6, which were found to be around the C > T, are indicated by a dashed circle. **c** The methylation level of the PCSK2 genomic region gene±2Kb is shown on the Y-axis

(chr20:17204815-17464832). The X-axis indicates the distance from the transcription start site (TSS, open arrow head). The position of the mutation detected only in nosn_6 is indicated by red arrowhead. The hypomethylated region of approximately 0.5 kb around the C > T mutation is indicated by the dashed line. **d** Frequency of CpG C > T mutations within the CpG island (CGI) and CGI shore. CpG C > T mutations within the CpG island (CGI) or CGI shore of iPSC lines (*n* = 75 biological replicates) and germline (*n* = 816) are shown. The Y-axis indicates the number of CpG C > Ts in the CGI and CGI shore regions (±5 kbp) per cell line (iPSC) or individual (germline). An error bar shows the SD of the mean. **e** Distance of CpG C > T mutations from the nearest transcription start site (TSS). The left Y-axis indicates the number of CpG C>Ts per cell line (iPSC) or individual (germline). The right Y-axis indicates the number of mutations per iPSC when the number of germline mutations is set to 1. Source data are provided as a Source Data file.

were also elevated in number. The most well-known mechanism underlying the C to T transition to date is the deamination of methylated cytosines. In addition, uracil is produced by the deamination of unmethylated cytosines, forms a mismatch with guanine, U/G. The

mismatch is however repaired very efficiently and rarely leads to mutations[34]. Therefore, the result is surprising due to unusual frequency of occurrence, and the mutation of C to T in sequences other than CpG showed a biphasic nature, indicating an occurrence at an

unusually high rate in lines with a high total number of mutations (Fig. 4, Supplementary Figs. 5 and 12).

Of note mutations in CpG sequences are of particular biological importance compared to those in other nucleotides, since CpG sequences are substrates for DNA methylation and thus play a central role in epigenomic regulation. They are also the recognition sites for factors with CpG-specific binding ability that regulate chromosome structure and function (such as Polycomb). The medical use of iPSCs thus requires careful analysis, not only of CDSs, but also of mutations that occur at CpG sites in regions other than CDSs. Further to this, our analysis confirmed the presence of many mutations in retrotransposons, particularly AluY. An elevated number of C > T mutations in retrotransposons may indicate their activation in iPSCs, as mentioned above. On the other hand, it has been reported previously that inadequate suppression of retrotransposons in iPSCs can affect their differentiation[35]. It has also been reported that differentiated cells (such as brain cells) in which retrotransposons are not fully suppressed are more likely to develop neurodegenerative diseases[36].

These results raise the question of the extent to which genomic mutations that occur during the generation of iPSCs are a problem that should be controlled for quality and safety in regenerative medicine. Evaluations of only the coding regions of oncogenes might not be sufficient[37] in this context, and sequencing of the entire genome, including repetitive sequences, required. The recent advent of single-molecule long-read sequencing has brought us closer to solving this difficult problem[38]. However, most mutations that occur in only one allele are not expected to have acute biological effects. Furthermore, cell technology must be developed to eliminate abnormal iPSC-derived cells from the body (suicide switch)[39]. Given these circumstances, it would not be impossible to use iPS cells for medical treatment if they do not have mutations that are clearly predicted to be deleterious and provided that careful follow-up is performed with an awareness of the mutations contained in the entire genome of these cells.

There will be no substitute going forward for generating iPSCs with as low a total number of mutations as possible, and erythroblast-derived iPSCs are attractive in this regard. These particular iPSCs contain so few mutations that it is possible to establish cell lines that are completely free of those that are potentially problematic[10]. However, one important issue remains to be addressed for all iPSCs, which is the problem of retrotransposons. The dynamics of retrotransposons have not been well understood to date due to the technical limitations of existing analytical methods. Recently however, it has become possible to detect them with higher sensitivity, and this has revealed that they may be active in iPSCs[27]. Not only does retrotransposition have the potential to significantly alter genome structure, any iPSCs with incomplete retrotransposon silencing and differentiated cells derived from them could potentially have various functional problems as mentioned above. This will have a very direct impact on the future decision as to whether iPSCs can be used in medicine.

In active DNA demethylation, methylated cytosine (5mC) is oxidized to a hydroxymethyl (5hmC), formylmethyl (5fC), and carboxymethyl cytosine (5caC), which are then recognized by glycosylase and replaced by unmethylated cytosine by the BER (base excision repair) reaction. However, such recruitment of the DNA repair pathway harbors a risk of mutagenesis. In fact, in vitro and in vivo DNA replication experiments using plasmids containing synthetic 5fC and 5caC as substrates have suggested that these modified 5mC bases have the potential to cause mutations[40–42]. However, 5fC and 5caC are thought to be efficiently recognized and eliminated by glycosylases, in contrast to 5hmC. It remains to be seen therefore whether these two modified bases actually cause mutations in cells with normal DNA repair activity under the control of DNA demethylation reactions.

We have previously reported that the rate of cell divisions at the very early stages (days 2-3) of iPSC generation is markedly higher (-7 h/cell division) and, interestingly, the cells at this stage are more tolerant to radiation[10,43]. Our further investigations revealed that this period does not cause cell cycle arrest, resulting in a transient reduction in DNA repair activity, and that this phenomenon is universally observed in a variety of iPSC generation methods in both mice and humans[10]. This result led us to infer that various base modifications arising during genome reprogramming escape repair and undergo subsequent DNA replication, leading to mutations. Indeed, the generation of 8-oxoG by ROS exposure leads to a high rate of C > A/G > T mutations during iPSC generation. Furthermore, our current analysis has indicated that a large number of C > T transitions occur at non-CpG sites, whereas uracil, which is produced by deamination of cytosine and is not present in DNA, is very efficiently recognized by glycosylases and eliminated by DNA repair mechanisms. This result also supports our contention and implies that the intermediate products of DNA demethylation, 5fC and 5caC, can also be mutated under conditions of genome reprogramming.

In this present study, we suggest a close relationship between DNA demethylation and CpG C > T mutagenesis, but we cannot show definitively that mutations in iPSCs are actually caused by this mechanism. Another question for the future will be this causative factor as the increase in CpG C > T mutations seen upon the forced expression of *Tet1* was not evident with the forced expression of *Tet2* (Supplementary Fig. 6b).

In conclusion, although CpG-specific C > T mutagenesis has been identified during iPSC generation in addition to cytosine deamination, its causative factors remain unclear. Our forced expression experiment seemed to mimic the mutation profiles observed in iPSCs, but further extensive studies are needed to elucidate the physiological causative factors of the transitions in genome reprogramming. It remains possible that other factors than Tet1 play this role.

## Methods

### Human iPSCs used in the mutation analyses

The following previously reported WGS data were used for SNV analysis: three iPSC lines using retroviral vectors (R-HDFa)[10], 5 iPSC lines using episomal vectors (pCE-hOCT3/4, pCE-SK, pCE-hUL, pCE-mp53DD and pCXB-EBNA1)[11] and 14 iPS cell lines established from enriched erythroblasts from cord blood using an all-in one episomal vector (CB-epi iPSCs)[10]. For the establishment of R-HDFa-iPSCs and E-HDFa-iPSCs, we used the same lot and passage stock of human dermal fibroblasts (HDFa, FC-0024; Lifeline Cell Technology, Frederick, MD, USA).

We also evaluated 53 iPSC lines established from human dermal fibroblasts using the Sendai virus vector by the HipSci project (http://www.hipsci.org/)[17]. From the HipSci database, 50 clones meeting the following criteria were selected for this analysis: iPSCs established from dermal fibroblasts of a healthy donor; total bases sequenced >120 Gb; Pluritest pluripotency score ≥15, and the availability of sequence reads for the somatic cells and the same somatic cell-derived sister clone. In addition, one set of sister iPSC lines was also analyzed, in which 3 sister clones (xomm-1, 2 and 3) from single donor were available, unlike the other 50 cell lines (only 2 sister lines were available): total bases sequenced ≥80 Gb; Pluritest pluripotency score ≥40.

### Germline mutations

Large-scale autosomal de novo mutation analyses of newborns and their parents have been reported previously, and we used these published data as germline mutations[44]. These data included mutations obtained as a result of the analysis of 816 individuals, but they were not reported as individual information. For this reason, statistical processing such as inter- individual variation was not possible.

## Animals

C57BL/6 (Japan SLC), Nanog-GFP tg [STOCK Tg (Nanog-GFP, puro) 1Yam] (RBRC#02290)[45], and B6.Cg-Trp53<tm1Sia >/Rbrc (RBRC01361)[46] were used for MEF preparation. All animal procedures were approved by the Ethics Committee of the National Institutes for Quantum Science and Technology.

## Plasmid Construction and mouse iPSCs generation

pMXs- Oct3/4, -Sox2, -Klf4 and -c-Myc were gifts from Shinya Yamanaka (Addgene plasmid # 13366, #13367, #13370 and #13375)[1]. pcDNA3-Tet1 and pcDNA3-Tet2 were gifts from Yi Zhang (Addgene plasmid # 60938, #60939)[47]. An ApaI-ApaI fragment (6.2Kb) containing the *Tet1* ORF fragment, and for *Tet2*, ApaI-BstPI and BstPI-XhoI fragments (total 6.2Kb) containing the *Tet2* ORF fragment were prepared from the pcDNA3-Tet1 and pcDNA3-Tet2, respectively and inserted into the retroviral vector pMXs. AAV shRNA vectors against mouse *Tet1* and *Tet2* were donated by Hongjun Song (Addgene plasmid #85742 and #85743)[48], and their MfeI-XbaI fragments were inserted into pMXs. MEFs were isolated from E13.5 embryos of male C57BL/6 mice, and grown in DMEM supplemented with 10% FCS. pMX vectors containing Oct3/4, Sox2, Klf4, c-Myc and an additional constructs (Tet1, Tet2, shRNA-Tet1 or shRNA-Tet2) were transfected into PlatE cells using FuGENE 6 (#E2691, Promega, Madison, WI, USA)[1]. The collected supernatant was filtered, supplemented with polybrene (4 μg/ml), and added to C57BL/6 single embryo derived fibroblasts 2 (MEF2) or MEF24 in 6 well-plates at $6.75 \times 10^4$ cells/well. The cultures were incubated for 16 h to facilitate infection and the medium was then replaced with DMEM supplemented with 15% fetal calf serum, leukemia inhibiting factor (#ESG1106, Merck, Darmstadt, Germany), penicillin/streptomycin, L-glutamine, nonessential amino acids, and 2-mercaptoethanol (2-ME). The 15% FCS supplement was subsequently replaced with 15% knockout serum replacement (#10828028, Invitrogen, Carlsbad, CA, USA) at day 4 post-infection. Genomic DNA was prepared from the iPSCs and analyzed by PCR to confirm integration of the *Tet1* and *c-Myc* vectors. shRNA sequences and primers used in this study are shown in Supplementary Table 4[47,49].

## Mutation analyses of mouse iPSC cell lines

We also re-analyzed the SNV profile of C57BL/6J single embryonic fibroblasts (MEF5)-derived iPS cells that we had previously established using retrovirus vectors under various conditions[10]. Two ntES cell lines established from the MEF5 were also used[16].

## Whole genome sequencing and identification of SNVs

Genomic DNA was prepared from iPSCs and parental somatic cells using a DNeasy Blood & Tissue kit (Qiagen, Hilden, Germany), and sequencing libraries were then constructed using the TruSeq DNA PCR free library prep kit (Illumina, San Diego, CA, USA) or the MGIEasy PCR-Free DNA Library Prep kit (MGI, Shenzhen, China). WGS was performed using a Hiseq X, Novaseq6000 (Illumina) or DNBseq-T7 sequencer (BGI, Shenzhen, China) to obtain 120–140 Gbases per sample (151bp-pair ends). The reads were then mapped to the human reference genome (GRCh37/hg19) or to the C57BL/6 mouse reference genome (NCBI37/mm9). Mismatches of 3% (length fraction, 1; similarity fraction, 0.97) for human iPSCs or 2% (length fraction, 1; similarity fraction, 0.98) for C57BL/6 mouse iPSCs were allowed and mapped, and only reads that were uniquely mapped were used. SNV candidates were identified using a CLC Genomics Workbench (Qiagen). The following parameters were applied: minimum quality of central base, 30; minimum average quality of surrounding bases, 15; window length, 11; maximum gap and mismatch count, 2; and depth ≥20. The following SNV candidates were excluded: variants also detected in parental cells (number of variant alleles in parental cells ≥2); known variants (dbSNP128 or dbSNP150 common SNP); variants also detected in sister clones (number of variant alleles in sister clones ≥2); SNVs in the simple repeat and homopolymeric regions. Finally, regions that appeared to be mapping or alignment errors were visually excluded, then SNVs with variant allele frequencies of 35-65% were identified. In the analysis of MEF5-derived iPSCs and ntESCs, the SNVs detected in sister clones were not excluded to identify shared SNVs. For the identification of retrotransposons in the genome we used RepeatMasker[50].

## DNA methylation analysis of iPS cells

DNA methylation profiling data of HipSci iPSCs using Infinium HumanMethylation 450 BeadChip (Illumina) are available from EMBL-EBI (E-MTAB-4059). We analyzed the methylation levels at the CpG sites near the CpG C > T mutations[51].

## RT-qPCR to assess *TET1* gene expression

Total RNA fractions were prepared using the RNeasy Mini Kit (Qiagen), and reverse transcription was performed using SuperScript III First-Strand Synthesis System (Thermo Fisher Scientific, Carlsbad, CA, USA), followed by real-time PCR analysis using TB Green Premix Ex Taq II (Takara Bio, Kusatsu, Japan) with the StepOnePlus Real-Time PCR System (Thermo Fisher Scientific). Data were normalized to *GAPDH* gene expression levels. PCR primer sequences and the cells used in this study are presented in Supplementary Tables 4 and 5.

## RNA-seq for expression analysis of SINE transcripts

RNAseq libraries were prepared using the Illumina TruSeq Stranded mRNA Sample Preparation Kit and then analyzed by paired-end sequencing (101 bp) on the Illumina NovaSeq X sequencer. The resulting sequence reads were mapped to the reference human genome (GRCh37/hg19) using STAR aligner version 2.7.10b[52]. The mapped data were then analyzed using TEtranscripts for SINE transcript quantification[53] and the count data were subsequently normalized using DESeq2 version 1.42.1[54].

## Distribution of CpG C > T mutations in DMR and other genomic regions

DMRs across 14 tumor types (TCGA dataset) have been reported previously[26]. We focused on the regions defined as a DMR in at least 7 of the 14 tumor types. CpG C > T and non-CpG C > T mutations identified in iPSCs were compared in DMRs and other genomic regions. These regions were normalized by CpG or other C abundance.

## Point mutation analysis using p53−/− MEFs

p53 +/− mice were mated and single embryo-derived p53−/− MEFs were prepared from E13.5 male embryos. Two p53−/− MEFs (MEF5 and 16) were used. They were cultured in 10% FCS supplemented DMEM and infected with pMXs-Tet1 or pMXs (mock). Single cell isolation was performed three days later; for each condition, 1 cell/ml cell solution was seeded at 500 μl/well (equivalent to 0.5 cells/well) in a 48-well plate ×10 (480 total wells). After 30 days of culture, cells were harvested from wells in which cells had successfully proliferated. A second single cell isolation was then performed to expand to the 500–700 cells required for WGS. Whole genome amplification was performed for WGS from these 500–700 cells using the REPLI-g Advanced DNA Single Cell Kit (Qiagen). Integration of these constructs was confirmed by PCR analysis.

## Reporting summary

Further information on research design is available in the Nature Portfolio Reporting Summary linked to this article.

# Data availability

Raw sequencing data of WGS generated in this study have been deposited in the DDBJ Sequence Read Archive (DRA) under accession codes DRA015842 and DRA017394-017396 [https://ddbj.nig.ac.jp/resource/sra-submission/DRA017394], [https://ddbj.nig.ac.jp/resource/sra-submission/DRA017395], [https://ddbj.nig.ac.jp/

resource/sra-submission/DRA017396]. Raw sequencing data of RNA-seq generated in this study have been deposited in the DRA under accession codes DRA018316 (DRR540229-DRR540235) [https://ddbj.nig.ac.jp/resource/sra-submission/DRA018316]. The previously generated raw sequencing reads of WGS used in this study are available in the DRA under accession codes DRA002956, DRA006232, DRA006457, DRA006622, DRA007325, DRA007336, DRA008453, DRA008459 and DRA012278. All materials and data are available upon request from the corresponding authors. Source data are provided with this paper.

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

## Acknowledgements

We thank Yi Zhang (University of North Carolina) for providing the *Tet1* and *Tet2* genes via Addgene, S. Yamanaka (Kyoto University) for providing the OSKC genes via Addgene and Nanog-EGFP mice via RIKEN BRC, H. Song (Johns Hopkins University) for providing shRNA-Tet1 and shRNA-Tet2 constructs via Addgene, T. Kitamura (University of Tokyo) for providing PlatE cells. We thank K. Kadota (University of Tokyo) and Y. Kasama (Visual Technology, Inc.) for their supporting bioinformatics analysis. We also thank Oohata T (Yokogawa Electric Corporation) and H. Yoshida (National Institutes for Quantum Science and Technology) for technical assistance. This study makes use of data generated by the HipSci Consortium, funded by The Wellcome Trust and the MRC. This work was supported in part by the JSPS KAKENHI grant number 21H02689 (to R.A.) and AMED grant number JP23zf0127008 (to M.A.).

## Author contributions

R.A. and M.A. conceived the study and wrote the manuscript. R.A., Y.H. and M.A. designed the experiments. Y.H., K.I., M.S., M.F. and R.A. contributed to the establishment and subsequent evaluation of the iPS cell lines. T.S. performed bioinformatics data analyses. S.K. and K.I. performed the nanopipette experiments.

## Competing interests

The authors declare no competing interests.
