## [Peer Review File · Nature Communications]

iPS cell generation-associated point mutations include many C>T substitutions via different cytosine modification mechanismsREVIEWER COMMENTS

Reviewer #1 (Remarks to the Author):

In this study entitled 'DNA demethylation during iPS cell generation causes an elevated C to T transition frequency', authors show that the genomic mutations in iPSCs are de novo events and the DNA methylation sequence CpG is a key mutagenic site. Additionally, the authors also show that demethyltransferase Tet1 increases the frequency of C to T transition during iPS cell generation. Since multiple previous studies have suggested that human ES and iPS cell lines have a higher number of genetic mutations than normal cells, a careful understanding of the genomic instability of individual human ES and iPS cell lines is important for future therapeutic applications in regenerative medicine. In this regard, the manuscript seems timely and important. However, there are multiple problems with data presented and analysis as described below. As a result, some conclusions of the manuscript are not strongly supported by the presented data.

Major comments:

1. This study claims that the origin of mutations identified in iPS cell genomes occurs during the initialization phase. Have you confirmed a relationship between the different establishment methods and the frequency of CpG methylation and the expression level of TET1? If there is a correlation with expression levels, it would be important data to suggest that TET1 acts on mutagenicity during iPS cell establishment.
2. In the current study, we found that the frequency of C to T transitions was increased in iPSC cell lines generated with OSKC + Tet1 compared to OSKC-only controls. Based on this observation, they argued that DNA demethylation at CpG sequences by Tet1 is responsible for the C to T transition. However, data on DNA demethylation and genomic variation in the case of suppression of Tet1 expression have not been presented. At the very least, it would be necessary to determine whether suppression of Tet1 expression reduces the frequency of C to T transitions.
3. Furthermore, several points were felt to be missing from the discussion:
 - From the viewpoint of quality and safety of raw materials used for regenerative medicine, to what extent are genomic mutations that occur during the establishment of iPS cells a problem that should be controlled?
 - Quality of sequence data (Depth and number of cases, etc.) required for mutation analysis on this scale.
4. The manuscript is well written and we think it could be of interest to different disciplines. However, we think it is technical. I think the readership would be broadened if you could describe the mutation analysis methods in more detail, especially using parent somatic cell ref seq data, so that it is easy to read for those who are not experts in the field.

Reviewer #2 (Remarks to the Author):

In the manuscript entitled "DNA demethylation during iPS cell generation causes an elevated C to T transition frequency" the authors investigate the mechanism leading to genome instability and mutations during reprogramming and generation of iPSCs. They first demonstrated that the majority of SNVs detected in iPSCs are not pre-existing in the parent somatic cell but originated during the reprogramming process. They then conducted mutation analysis on several human iPSC lines and found CpG as prominent mutation-prone site with a high C to T transition and a specific distribution in intergenic regions and in short interspersed nuclear elements (SINEs) subfamily AluY. Finally, they observed that the forced expression of Tet1 (together with Yamanaka factors) during reprogramming results in an increased C to T mutation frequency in iPSCs, specifically at CpG

sequences.

This study is interesting however the authors need to address the following issues:

1. Is Tet1 overexpression introducing C>T mutations only during reprogramming or is it a general phenomenon?
2. Do C>T mutations have a biological relevance? For example, impaired differentiation of iPSCs, impaired potential of iPSCs?

General comment:

I would suggest improving the writing throughout the manuscript as it is sometimes unclear and some concepts are reiterated.

Reviewer #1 (Remarks to the Author):

In this study entitled 'DNA demethylation during iPS cell generation causes an elevated C to T transition frequency', authors show that the genomic mutations in iPSCs are de novo events and the DNA methylation sequence CpG is a key mutagenic site. Additionally, the authors also show that demethyltransferase Tet1 increases the frequency of C to T transition during iPS cell generation. Since multiple previous studies have suggested that human ES and iPS cell lines have a higher number of genetic mutations than normal cells, a careful understanding of the genomic instability of individual human ES and iPS cell lines is important for future therapeutic applications in regenerative medicine. In this regard, the manuscript seems timely and important. However, there are multiple problems with data presented and analysis as described below. As a result, some conclusions of the manuscript are not strongly supported by the presented data.

Major comments:

1. This study claims that the origin of mutations identified in iPS cell genomes occurs during the initialization phase. Have you confirmed a relationship between the different establishment methods and the frequency of CpG methylation and the expression level of TET1? If there is a correlation with expression levels, it would be important data to suggest that TET1 acts on mutagenicity during iPS cell establishment.
2. In the current study, we found that the frequency of C to T transitions was increased in iPSC cell lines generated with OSKC + Tet1 compared to OSKC-only controls. Based on this observation, they argued that DNA demethylation at CpG sequences by Tet1 is responsible for the C to T transition. However, data on DNA demethylation and genomic variation in the case of suppression of Tet1 expression have not been presented. At the very least, it would be necessary to determine whether suppression of Tet1 expression reduces the frequency of C to T transitions.

Response:

We thank this reviewer for these comments on the core issues underpinning our current study. Since comments 1 and 2 seem to be closely related, we have combined our responses to both below.

We wanted to perform Tet suppression experiments, but it has been reported that either Tet KO or shRNA suppression of this gene inhibits genome reprogramming itself and thus reduces the efficiency of iPSC generation^{1,2}. Hence, even if iPSCs were generated under Tet suppression conditions, they would likely be incomplete or enriched with iPSCs in which suppression did not work properly. We did not perform these experiments

therefore. In addition, since the Tet family consists of three paralogs, Tet1-3, there was a concern that even if one paralog was suppressed, it would likely be complemented by another³.

However, the point the reviewer has made in this regard is obviously very important and we thus performed suppression experiments not only on Tet1, but also on Tet2 and both Tet1 and -2. However, the efficiency of iPSC generation was low (Supplementary Fig. 5a), and the generated iPSCs showed a distorted morphology that differed from normal iPSCs, especially at the beginning of their generation. SNV analysis of these distorted iPSCs showed no difference from the controls (Supplementary Fig. 5b).

In our experiments, we first obtained data suggesting that there is mutagenic potential in the DNA demethylation reaction during genome reprogramming via the forced expression of the Tet1 gene (in the revised manuscript, to make the results even more robust, we added two additional Tet1 transgenic iPSCs that were not available at the time of the first submission, for a total of six iPSC lines; Fig. 5b-g). In addition, we also observed a clear increase in SINE mutations (Fig. 5g). This result seemed to reproduce the phenomenon observed in the mutation analysis of the iPSCs genome, i.e. increased mutations in AluY retrotransposons. However, as the reviewer points out, it cannot be concluded from the Tet1 forced expression experiment alone that the mutations in iPSCs include those caused by DNA demethylation.

Due to the inability to conduct Tet suppression experiments, we could only use a nonreverse genetic approach. We thereby explored a series of observations that strongly suggested a close relationship between the DNA demethylation reaction and the C>T mutation. An interesting finding revealed such a close relationship between the two. When the association between the number of C>T mutations and the total number of mutations was examined separately for CpG and non-CpG sites (i.e. the origin sequence where the mutation occurred), a contrasting correlation was revealed in which the percentage of C to T transitions at non-CpG sites showed a strong positive correlation ($r=0.81$) with the total number of SNVs, whereas the percentage of mutations occurring at CpG sites showed a weak negative correlation ($r=-0.29$) (Fig. 6a and b). Moreover, in

iPS lines with significantly lower total SNV counts, the ratio of C>T transitions at CpG sites (C to T transitions occurring at the CpG site) to all mutations was found to be significantly higher (Fig. 4a).

Given that non-CpG C>T transitions are thought to be due to deamination, we hypothesized that a mechanism other than deamination might be at work in iPSCs with such a low total number of mutations, where a C>T aberration at CpG sites is common, but at non-CpG sites is rare. We focused our analysis in this regard on iPSCs with few mutations, especially those derived from erythroblasts. We compared the mutation rates in DMRs (differentially methylated regions) and other genomic regions and found that CpG C>T mutations in DMRs occurred more frequently in these iPSCs with fewer mutations (Fig. 6c). A similar trend was clearly observed in HDF-derived iPSCs, the second least mutated iPSCs after erythroblast-derived iPSCs, although was weaker than in erythroblast-derived iPSCs. In addition, the expression of the Tet1 gene in the parental somatic cells showed a similar order, i.e. erythroblasts > HDF (Fig. 6d).

In addition to these observations, we believe it is important to note that the increase of CpG C>T mutations in retrotransposons, especially SINEs, was striking in iPSCs. This is because it has recently been shown that retrotransposition is activated upon iPSC reprogramming⁴, and DNA demethylation is the first step in this activation.

These observations collectively suggest that in genome reprogramming, DNA demethylation reactions are closely related to CpG C>T mutations, in addition to the previously known deamination reactions. Because of the biological significance of this, further analysis and more direct evidence is needed from future studies. (page8 line15 to page9 line6 and page9 line24 to page11 line2)

Furthermore, the following text summarizes at the end of the revised manuscript what is clear and what is not clear in this study.

‘In this present study, we found that DNA demethylation during iPSC generation has mutagenic potential. Furthermore, although we were able from our findings to suggest a

close relationship between DNA demethylation and CpG C>T mutagenesis, we were not able to show definitively that mutations in iPSCs are actually caused by this mechanism. Another question for the future will be this causative factor as the increase in CpG C>T mutations seen upon the forced expression of Tet1 was not evident with the forced expression of Tet2 (Supplementary Fig. 5b). In iPSC generation, Tet2 is expressed earlier in the first few days, whereas Tet1 induction is never high initially, but gradually increases over a period of about 10 days ^{2,5}. Many of the mutations identified in iPSCs are found in most of the cells that make up their colonies; this means that the mutations occur very early in their generation ⁶. It is possible that a forced expression of Tet2 may not have much effect where there is already a sufficient amount of endogenous protein. On the other hand, Tet1 may be more effective. This may also mean that there are differences in the DNA demethylation sites of Tet1 and Tet2. The Tet family is likely to be the causative factor of the phenomenon we have uncovered, but how it actually functions during genome reprogramming is a question that needs to be clarified in the future. On the other hand, elucidating the relationship between DNA demethylation and CpG C>T mutations in biological phenomena other than genome reprogramming is an important issue that may provide new insights into the mechanisms of carcinogenesis and other diseases.’

3. Furthermore, several points were felt to be missing from the discussion:

- From the viewpoint of quality and safety of raw materials used for regenerative medicine, to what extent are genomic mutations that occur during the establishment of iPSC cells a problem that should be controlled?

Response:

We thank the reviewer for these insightful comments and accordingly have added the following text to the Discussion:

‘Our current analysis of human iPSCs has revealed a much larger variation in the number of mutations than previously thought and also showed that CpG C>T mutations, which were the focus of this study, require further attention with respect to their effects on regulatory regions in the genome. These results raise the question of the extent to which genomic mutations that occur during the generation of iPSCs are a problem that should

be controlled from the point of view of the quality and safety of raw materials used for regenerative medicine. Evaluations of only the coding regions of oncogenes are certainly not sufficient ⁷ in this context, and sequencing of the entire genome, including repetitive sequences, will be required. The recent advent of single-molecule long-read sequencing has brought us closer to solving this difficult problem ⁸. On the other hand, most CpG C>T mutations that occur in only one allele are not expected to have acute biological effects. Furthermore, cell technology must be developed to eliminate abnormal iPSC-derived cells from the body (suicide switch) ⁹. Given these circumstances, it is likely that iPSCs could be used for medical treatment if they do not have mutations that are clearly predicted to be deleterious, and provided that careful follow-up is performed with an awareness of the mutations contained in the entire genome of these cells.

There will be no substitute going forward for generating iPSCs with as low a total number of mutations as possible, and erythroblast-derived iPSCs are attractive in this regard. These particular iPSCs contain so few mutations that it is possible to establish cell lines that are completely free of those that are potentially problematic ¹⁰. However, one important issue remains to be addressed for all iPSCs, which is the problem of retrotransposons. The dynamics of transposons have not been well understood to date due to the technical limitations of existing analytical methods. Recently however, it has become possible to detect them with higher sensitivity, and this has revealed that they may be active in iPSCs ⁴. Not only does retrotransposition have the potential to significantly alter genome structure, any iPSCs with incomplete retrotransposon silencing and differentiated cells derived from them could potentially have various functional problems. This will have a very direct impact on the future decision as to whether iPSCs can be used in medicine.’

- Quality of sequence data (Depth and number of cases, etc.) required for mutation analysis on this scale.

The total number of nucleotides sequenced, average depth, and genome coverage (≥ 20 reads) for each clone are shown in Supplementary Data 1.

4. The manuscript is well written and we think it could be of interest to different disciplines. However, we think it is technical. I think the readership would be broadened if you could describe the mutation analysis methods in more detail, especially using parent somatic cell ref seq data, so that it is easy to read for those who are not experts in the field.

Response:

We thank the reviewer for this suggestion. Unlike transcriptome analysis, which allows single cell analysis, it is still impractical to perform whole genome sequencing on single cells. Hence, WGS is used to analyze cell populations, but sequence heterogeneity within a population will arise during the growth process. We recognize that the issues that arise in this process will complicate the interpretation of the data. To facilitate the reader's understanding, we have provided a diagram of our iPSC mutation analysis framework (Supplementary Fig. 1).

Reviewer #2 (Remarks to the Author):

In the manuscript entitled “DNA demethylation during iPSC cell generation causes an elevated C to T transition frequency” the authors investigate the mechanism leading to genome instability and mutations during reprogramming and generation of iPSCs. They first demonstrated that the majority of SNVs detected in iPSCs are not pre-existing in the parent somatic cell but originated during the reprogramming process. They then conducted mutation analysis on several human iPSC lines and found CpG as prominent mutation-prone site with a high C to T transition and a specific distribution in intergenic regions and in short interspersed nuclear elements (SINEs) subfamily AluY. Finally, they observed that the forced expression of Tet1 (together with Yamanaka factors) during reprogramming results in an increased C to T mutation frequency in iPSCs, specifically at CpG sequences.

This study is interesting however the authors need to address the following issues:

1. Is Tet1 overexpression introducing C>T mutations only during reprogramming or is it a general phenomenon?

Response:

We thank the reviewer for these insightful comments on this very important and interesting point. We have shown that DNA demethylation is mutagenic in the context of genome reprogramming, and we also believe that a very important scientific question is whether DNA demethylation is also mutagenic in other biological phenomena. We hope that our present report will further clarify the relationship between the epigenome and the genome. We have added the following text to the Discussion to provide clarity:

‘Whole genome mutation analysis by WGS is limited to cells that can undergo clonal expansion, as single cell WGS is not yet available. This method can therefore only be employed with iPSCs, tumors, and a very limited number of cell types that can be expanded sufficiently from a single cell. WGS analysis of a Tet1-transfected cell population cannot provide valid data because this cell population is very heterogeneous in terms of mutations. To accurately study the effect of the Tet1 gene on mutagenesis therefore, it will be necessary to isolate a single transgenic cell from a post-transfection cell population, and this requires a system that can culture a single cell to a sufficient population size. However, it is well known that single cell expansion from primary cell

populations is not possible in mice (and very few human cell types are capable of expansion from single cells to WGS-enabling levels).

Since we have found that even primary cultured mouse cells maintain their proliferation after single cell isolation when p53 is deleted (manuscript in preparation), we decided to use p53^{-/-} MEFs in this experiment. On the other hand, as mentioned above, introducing the Tet1 gene directly into a single cell was a theoretically possible approach, but was abandoned when it was found that doing so in a primary cultured cell population resulted in cell growth arrest (Supplementary Fig. 7).

We used p53KO-MEFs (5 and 16) to compare Tet1-transduced and untransduced (control) cells (Supplementary Fig. 9a, b). Briefly, the cells were infected with a retroviral vector containing Tet1 gene, and single cell isolation was performed three days later. The cells were then allowed to grow for one month (and some were set aside for WGS). After this, single cell isolation was performed again, and the number of cells was allowed to grow to 500-700 cell populations and subjected to WGS. In this system therefore, Tet1 was expressed intracellularly for one month to observe its effect. As a result, a trend toward increased CpG C>T mutations was observed in Tet1-transfected cells, although this was not statistically significant (Supplementary Fig. 9c). These observations suggested that mutations may also occur during biological events other than genome reprogramming, such as carcinogenesis following deletion of the p53 gene, and regeneration events in amphibians, where a transient decrease in p53 activity occurs¹¹. This sheds new light on the molecular mechanisms underlying these important biological phenomena. In our present experiments, we could only use p53^{-/-} cells due to technical limitations, but addressing the question of whether DNA demethylation produces mutations even under normal p53 conditions is definitely critical, as it may be strongly associated with development and differentiation.'

2. Do C>T mutations have a biological relevance? For example, impaired differentiation of iPSCs, impaired potential of iPSCs?

Response:

We apologize for the lack of information on these important aspects of mutation research. We discuss the biological effects of the single nucleotide substitution C>T (occurring at the CpG site) and examined each of the CpG C>T mutations detected in the iPSCs genome in our study. Four possible mechanisms were considered: 1) mutations in the CDS (described in the first version of the paper), 2) mutations affecting splicing, 3) mutations in genomic regions other than the CDS, such as intronic and intergenic regions, and 4) mutations in the CGI. In the case of 2), no instances were found among the mutations detected in the iPSCs, but in the other three examples, these types of mutations were indeed confirmed.

In the case of 1), STOP codons were frequently observed: 105 of 168 mutations detected in CDSs caused amino acid substitutions, with missense and STOP codons accounting for 97 and 8 amino acid substitutions, respectively. In addition, STOP codons causing protein truncation accounted for 4.8% of the total CpG C>T mutations detected in CDSs. Next, the C>T transition at CpG sites to splicing in the peri-exon region is also known to have many effects, causing dramatic conformational changes in proteins such as deletions and exon skipping¹². For the mutations in 3) the question of whether CpG C>T mutations in genomic regions other than coding regions are associated with biological aberrations was investigated. A large number of CpG C>T changes have been reported as SNPs and are powerful tools for identifying genes or regions associated with disease (i.e. associated SNPs). In recent years, attempts have also been made to search for causal SNPs in disease. Alsheikh et al. conducted a comprehensive literature review of non-coding SNPs and reported that 309 SNPs had been experimentally validated (BMC Medical Genomics 2022). These included C>T mutations occurring at 52 CpG sites (Supplementary Table 3). Many of these are C>T mutations in transcriptional regulatory regions that alter transcription factor binding and reduce gene expression. They have also been shown to be risk factors for type 2 diabetes, Parkinson's disease, ischemic heart disease, etc. (Supplementary Table 3). Among the mutations detected in iPSCs in our present study, especially via sister clones analysis, we found a close relationship between mutation and methylation profile. We used sister clones to compare cells with similar methylation backgrounds to study the

effect of the CpG C>T mutation on methylation. As a result, an interesting pair of sister iPSC clones (nosn_1 and nosn_6) was detected; in this pair, only nosn_6, which has the CpG C>T mutation, showed decreased methylation levels around this mutation.

However, this result does not necessarily indicate that this mutation affects methylation. Therefore, we further examined the methylation levels in this region in the remaining 24 sister clone pairs and found that even in a small number of pairs (2/24 pairs), there were large differences in methylation levels between the sisters (all five hypomethylated CpG sites around the mutation detected in nosn_6 had a β value of 0.2 or greater). This result indicates that this region can be methylation variable even without the identified CpG C>T mutation, suggesting that the mutation is a consequence rather than a cause of the hypomethylation. On the other hand, the difference in methylation between the sisters is greatest between nosn_1 and nosn_6, so the effect of CpG C>T in such a region prone to methylation fluctuations is a question worth clarifying in the future (Fig. 7b and c). Thus, a C>T occurring at a single CpG site, even outside the coding region, has the potential to cause abnormal biological phenomena. Finally, regarding the mutations in 4), many CpG C>T aberrations were detected in iPSCs around the TSS region, where mutations are usually rare, and there is also a tendency for mutations to occur less frequently closer to the TSS region, which was not observed in iPSCs (Fig. 7d and e).

Mutations in CpG sequences are of particular biological importance compared to mutations in other nucleotides because CpG sequences are substrates for DNA methylation, which plays a central role in epigenomic regulation, or for factors with CpG-specific binding capacity that regulate chromosome structure and function, such as Polycomb. Factors with CpG sequence-specific binding ability are known to be involved in the regulation of various biological phenomena. In the medical application of iPSCs, careful analysis of not only CDSs but also mutations occurring at CpG sites in regions other than CDSs should be required.

In addition, we found the presence of many mutations in retrotransposons, especially AluY (Fig. 3c). This may indicate the activation of retrotransposition during iPSC generation, which was recently revealed, and suggests that demethylation of the retrotransposon locus, the first step of retrotransposition, may have caused the C>T

transitions. Moreover, it has been reported that the more inadequate the suppression of retrotransposons in iPSCs, the more problematic their differentiation¹³. Importantly, it has also been reported that differentiated brain cells in which retrotransposons are not fully suppressed are more likely to develop neurodegenerative diseases¹⁴.

In light of the above evidence, careful identification and handling of C>T transitions at CpG sites is necessary in any regenerative medicine applications that will use cells and tissues differentiated from iPSCs (page11 line4 '**Biological relevance of the C to T transition at CpG sites**').

General comment:

I would suggest improving the writing throughout the manuscript as it is sometimes unclear and some concepts are reiterated.

Response:

We apologize for the duplications and ambiguities in the first submitted version, which made it difficult to read. In particular, the discussion was badly repetitive. We have rewritten it carefully.

references

- 1 Doege, C. A. *et al.* Early-stage epigenetic modification during somatic cell reprogramming by Parp1 and Tet2. *Nature* **488**, 652-655 (2012).
- 2 Gao, Y. *et al.* Replacement of Oct4 by Tet1 during iPSC induction reveals an important role of DNA methylation and hydroxymethylation in reprogramming. *Cell Stem Cell* **12**, 453-469 (2013).
- 3 Hu, X. *et al.* Tet and TDG mediate DNA demethylation essential for mesenchymal-to-epithelial transition in somatic cell reprogramming. *Cell Stem Cell* **14**, 512-522 (2014).
- 4 Gerdes, P. *et al.* Retrotransposon instability dominates the acquired mutation landscape of mouse induced pluripotent stem cells. *Nat Commun* **13**, 7470 (2022).
- 5 Polo, J. M. *et al.* A molecular roadmap of reprogramming somatic cells into iPS cells. *Cell* **151**, 1617-1632 (2012).
- 6 Sugiura, M. *et al.* Induced pluripotent stem cell generation-associated point mutations arise during the initial stages of the conversion of these cells. *Stem*

- Cell Reports* **2**, 52-63 (2014).
- 7 Mandai, M. *et al.* Autologous Induced Stem-Cell-Derived Retinal Cells for
Macular Degeneration. *N Engl J Med* **376**, 1038-1046 (2017).
- 8 Logsdon, G. A., Vollger, M. R. & Eichler, E. E. Long-read human genome
sequencing and its applications. *Nat Rev Genet* **21**, 597-614 (2020).
- 9 de Luzy, I. R. *et al.* Human stem cells harboring a suicide gene improve
the safety and standardisation of neural transplants in Parkinsonian rats. *Nat
Commun* **12**, 3275 (2021).
- 10 Araki, R. *et al.* Genetic aberrations in iPSCs are introduced by a transient G1/S
cell cycle checkpoint deficiency. *Nat Commun* **11**, 197 (2020).
- 11 Yun, M. H., Gates, P. B. & Brockes, J. P. Regulation of p53 is critical for vertebrate
limb regeneration. *Proc Natl Acad Sci U S A* **110**, 17392-17397 (2013).
- 12 Roca, X. *et al.* Features of 5'-splice-site efficiency derived from disease-causing
mutations and comparative genomics. *Genome Res* **18**, 77-87 (2008).
- 13 Ohnuki, M. *et al.* Dynamic regulation of human endogenous retroviruses
mediates factor-induced reprogramming and differentiation potential. *Proc Natl
Acad Sci U S A* **111**, 12426-12431 (2014).
- 14 Takahashi, F. *et al.* Immune-mediated neurodegenerative trait provoked by
multimodal derepression of long-interspersed nuclear element-1. *iScience* **25**,
104278 (2022).

REVIEWER COMMENTS

Reviewer #1 (Remarks to the Author):

The revised manuscript addresses some of the issues raised in the first revision, but no differences in CpG C>T in Tet1 and Tet2 suppression experiments were found.

The authors use a non-reverse genetics approach to identify correlations between DNA demethylation reactions and C>T mutations.

Using erythroblast-derived iPS cells or HDF-derived iPS cells with few mutations, they show that they exhibit CpG C>T mutations in the DMR. However, the expression levels of the Tet gene in erythroblast-derived iPS cells or HDF-derived iPS cells have not been shown. Nor have data been shown for iPS cells established using other initialization methods.

I can't recommend the revised manuscript will be published in Nature Communications due to lack of the molecular mechanism.

Reviewer #2 (Remarks to the Author):

The authors have addressed my concerns and improved the manuscript with better articulation and additional data.

The study is comprehensive.

The updated manuscript could be published in Nature Communications.

Reviewer #3 (Remarks to the Author):

The manuscript entitled "DNA demethylation during iPS cell generation causes an elevated C to T transition frequency" provides important and timely data to interpret mutational profiles observed in iPS cell lines. The strength of the manuscript is a series of elegant WGS datasets which convincingly show that most mutations observed in iPS cell lines are not preexisting SNVs but rather de novo mutations acquired during establishment of the cell lines. They also demonstrate that different numbers of mutations are acquired depending on derivation method as well as starting material used, two important observations for the field.

The authors also observe an overrepresentation of C to T transitions in CG context and conclude that DNA demethylation processes, via TET enzymatic activity, cause this. This conclusion is supported by data showing that overexpression of TET1, but not TET2, results in a modest increase in CG cytosine to thymine mutations. There is no attempt to use overexpression of catalytic dead TET1 enzyme to directly test the hypothesis that DNA demethylation is a causal factor, however this was also not requested by reviewers. Upon reviewer 1 request, the authors have attempted to suppress TET activity, but this yielded inconclusive results because of inefficient iPS cell generation. The authors have furthermore, upon reviewer 1 request, included a limited set of expression data to correlate TET1 expression levels with number of CpG C to T transitions (Fig 6d – although this figure appears to be missing statistical significance). As such, the authors have attempted to provide answers to the reviewer 1 concerns – although with limited success.

I am concerned about the authors conclusion that TET-mediated DNA demethylation is causing the observed mutational pattern. Surprisingly, the authors do not cite literature which show that C to T transitions in a CG context is the most common mutation in cancer (and most likely explains the general underrepresentation of CG dinucleotides in mammalian genomes – hence this also occur in germ cells) and that the rate of spontaneous deamination of 5mC into T is four-fold higher than the rate of deamination of C into U (e.g. PMID: 23945592, PMID: 27183007). This should at least be discussed in the context of the current data to inform the reader. The authors also do not discuss the relationship between effects DNA replication/cell division and the observed pattern of germline and iPS

cell mutations. The enrichment of CG C to T transitions found in retrotransposons, especially AluY elements, is interesting but is not followed up on a functional level by examining expression of retrotransposons, TET1 colocalisation, or 5fC/5caC enrichment at these sites during iPS cell derivation.

In summary, the manuscript contains important advances in relation to mutational processes in iPS cell derivation and deserves publication to a wider audience. However, I do not think that the main conclusion of the paper (presented in the title) is adequately supported by the data presented.

Although the authors have included a statement in the discussion: "although we were able from our findings to suggest a close relationship between DNA demethylation and CpG C>T mutagenesis, we were not able to show definitively that mutations in iPSCs are actually caused by this mechanism", it is my opinion that the manuscript is not ready for publication in its current form, and I would recommend that it is rewritten to make less strong statements about a role of DNA demethylation in this context.